# For Whom Is Anti-Bullying Intervention Most Effective? The Role of Temperament

**DOI:** 10.3390/ijerph16030388

**Published:** 2019-01-30

**Authors:** Annalaura Nocentini, Benedetta Emanuela Palladino, Ersilia Menesini

**Affiliations:** Department of Educational Science and Psychology, University of Florence, Ital Via di San Salvi, 12, Complesso di San Salvi Padiglione 26, 50135 Florence, Italy; benedettaemanuela.palladino@unifi.it (B.E.P.); menesini@psico.unifi.it (E.M.)

**Keywords:** effectiveness, moderators, temperament, anti-bullying, subgroup analyses

## Abstract

Studying moderators of the effects of anti-bullying universal interventions is essential to elucidate what works for whom and to tailor more intensive, selective, and indicated programs which meet the needs of non-responders. The present study investigated whether early adolescents’ temperament—effortful control (EC), negative emotionality (NE), and positive emotionality (PE)—moderates the effects of the KiVa anti-bullying program. The sample consisted of 13 schools, with 1051 sixth-grade early adolescents (mean age = 10.93; SD = 0.501), randomly assigned to the KiVa intervention (seven schools; *n* = 536) or to the control condition (six schools; *n* = 516). Adolescents reported bullying and victimization before the intervention (pre-test) and after (post-test). Temperament was assessed by a self-report pre-test. Findings showed that EC and NE moderated intervention effects on bullying, indicating that subgroups with high levels of EC, and with low and medium levels of NE were those who benefited most from the intervention. The low-EC subgroup showed a lower increase compared to the control condition, with a considerable effect size. Conversely, the high-NE subgroup did not show any positive effects compared to the control group. Regarding victimization, findings showed that early adolescents with high and medium levels of PE were the subgroups who benefited the most from the intervention, whereas the low-PE subgroup was the most resistant. The present study confirms the relevance of considering temperament as a moderator of intervention effects, since interventions tailored to early adolescents with specific traits might yield larger effects.

## 1. Introduction

Evidence-based research about anti-bullying interventions needs to move beyond basic questions related to anti-bullying intervention efficacy. The mechanisms through which ameliorative effects may be exerted, the factors that alter the efficiency of the intervention within different subsamples, and the identification of those who are most likely to benefit from a given intervention are relevant research questions that need to be addressed. The modest effect size of reduction estimates found in studies on the efficacy of anti-bullying interventions, as well as the existing variability between studies and programs, suggests the need to pay more attention to potential moderators of the response to these interventions. Identifying children and adolescents who do not respond to interventions is, therefore, an important objective for tailoring more intensive selective and indicated programs which meet the needs of non-responders. The present study aimed to evaluate whether specific subgroups of early adolescents defined by their temperamental traits can be differently affected by a universal school-based anti-bullying program.

School bullying increasingly became a topic of both public concern and research efforts. The literature on the effectiveness of anti-bullying programs showed that they are effective in reducing bullying by 20% to 23%, and victimization by 17% to 20% [1]. These findings were substantially confirmed by the recent update by Gaffney and colleagues [2]. Overall, although these percentages correspond to a substantial amount of bullying prevention, the effect sizes are considered as small (standardized mean difference (d) values ranging from 0.14 to 0.17). There is still much to understand about why programs vary in effectiveness, and much to learn about improving prevention strategies to optimize the effect size of prevention programs.

Prevention science also recommends the testing of moderation models to better understand what works and for whom [3]. The importance of examining the consistency of intervention effects across subgroups was incorporated into the standards of evidence for identifying effective prevention programs adopted by the Society for Prevention Research [4,5]. In the literature on the effectiveness of anti-bullying interventions, only a few studies focused on uncovering these individual moderating factors, investigating the role of demographic characteristics (age and gender) [1,6,7], initial severity of behaviors and symptoms [7,8], type of involvement in either bullying or victimization or both (i.e., pure bullies, pure victims, or bully/victims) [9], and a general personality child trait called environmental sensitivity [10].

In relation to gender moderation, studies showed mixed results, with some suggesting that gender does not have a moderating effect on bullying and victimization decrease [7,11], and others reporting boys obtaining higher results for bullying reduction compared to girls [6,10]. The stronger positive effects on boys may be a consequence of boys’ initially higher scores on bullying and pro-bullying behavior that make them suitable targets for the intervention to reduce bullying, assisting, and reinforcing. Age is another variable showing contrasting moderating effects; greater reductions in bullying were documented for younger children compared to their older counterparts [12,13,14,15]. Other studies reported no differences between age groups [7], and others [1] asserted that greater intervention success can be attained when working with slightly older youths (age 11 or older). The initial severity of the behaviors was found to be a significant moderator of the effectiveness of anti-bullying programs. In agreement with the literature on other outcomes [8,16,17,18,19], those most involved in problematic behaviors at the baseline level were those benefiting the most from the intervention. The KiVa program was particularly effective in facilitating the perception of a caring school environment for the students who were most victimized before the intervention, and intervention effects on depression and self-esteem were strong only among the most victimized sixth graders [8]. In the study on the ViSC Social Competence Program [7], youths with initially high levels of aggressive behavior (including bullying) or victimization changed more in the respective variable compared with initially not-so-involved youths. Another interesting individual moderator included the involvement of children in both bullying and victimization roles. Specific subgroups of children (e.g., bullies, victims, bully/victims) followed a different trend after the KiVa intervention. Yang and Salmivalli [9] suggested that KiVa is effective in reducing the prevalence of bully/victims, and these effects are comparable or even larger than the effects on pure bullies and pure victims. Finally, an individual variable able to predict interindividual variability in the effectiveness estimates is a general personality trait called environmental sensitivity, which is defined as the inherent ability to perceive and process environmental stimuli [10]. A recent study showed that low-sensitivity children seem to be relatively resistant to the program’s effects in relation to reducing victimization, and low-sensitivity boys in relation to reducing internalizing behaviors. On the other side, highly sensitive children were more responsive to the anti-bullying program in relation to a decrease in victimization and on internalizing behaviors [10]. 

Environmental sensitivity may constitute a non-specific personality predictor for victimization. In relation to bullying, more specific personality or temperamental traits could play a key role in identifying children and adolescents who respond better or worse to the intervention.

However, studies are yet to investigate the moderating role of personality or temperament in relation to anti-bullying prevention programs, and very few investigated this role in the more general literature of externalizing behaviors. In particular, two studies conducted in the Netherlands showed that highly conscientious and highly agreeable juvenile delinquents [20], and highly conscientious and low extroverted children [21] benefited most from interventions targeted at reducing externalizing problems. These two studies showed that youth with a lower-risk personality profile for the development of delinquency benefited most from the intervention.

The association between personality traits and bullying and victimization was analyzed in a few studies [22,23,24,25,26,27,28,29]. Using the Eysenck Personality Inventory Junior, a study conducted in Ireland reported heightened levels of psychoticism and modest increases in extroversion and neuroticism among bullies [24]. Using the five-factor model, studies conducted in the United States (US) and in Italy confirmed that victimization is associated with higher levels of neuroticism, and lower levels of agreeableness and conscientiousness [22,26,28], and that bullying is associated with low agreeableness and low conscientiousness [22,23,26,28]. In the Italian studies, bullying was also associated with high levels of neuroticism [26,28]. Using the HEXACO Personality Inventory, Book et al. [23] found that bullying was significantly negatively correlated with traits of honesty–humility, emotionality, agreeableness, and conscientiousness. Finally, using the EATQ (Early Adolescent Temperament Questionnaire), Terranova and colleagues [29] showed the role played by fear reactivity and effortful control in bullying. Starting from these associations, we might expect differential effectiveness as a function of effortful control and negative emotionality in reducing bullying and victimization.

Given the considerable variability in anti-bullying intervention effects, additional research is required to identify the characteristics of the most responsive early adolescents. The issue of variation in effectiveness based on subgroup membership is particularly relevant for the development of the tiered framework in anti-bullying interventions. This model, including a universal, a selective, and an indicated tier [16], is essential for tailoring interventions according to the risk profile of subgroups of students.

Starting from these considerations, the current study evaluates whether the impact of the KiVa anti-bullying program varied as a function of early adolescents’ temperament defined using the three superordinate dimensions: effortful control (EC), negative emotionality (NE), and positive emotionality (PE). 

General recommendations for testing the moderation of the impact of an intervention proposed by Supplee et al. [30] and Wang et al. [31] were followed. Results were based first on the test of the heterogeneity of treatment effects, and secondly on subgroup analyses presented along with effect estimates within each level of each baseline moderator. Subgroups were specified using an empirical distribution: top 30%, medium 40%, and bottom 30%.

## 2. Methods

### 2.1. Participants

Participants of this study were part of a randomized controlled trial (RCT), which was not associated to an ethical code because an ethical committee was not established in our University until 2016. At that time, we were requested to strictly adhere to the ethical code of the Italian Association of Psychology. This RCT aimed at testing the effectiveness of the KiVa anti-bullying program in Italy, involving 13 comprehensive schools located in three cities in Tuscany. A detailed description of the trial design, and the recruitment and retention of participants in the RCT is reported in Nocentini and Menesini [12]. Data were collected in two waves: September–October 2013 (T1: pre-treatment) and May–June 2014 (T2: post-treatment). The sample included in this study is limited to secondary schools because the EATQ questionnaire is not adequate for primary school and was used only with the secondary subsample. In order to recruit children, parents were sent information letters together with a consent form. In total, 94% of the target sample provided active consent for study participation. Overall, 1052 early adolescents from grade 6 filled out the questionnaires at T1 and 984 at T2. The attrition is mainly explained by absences of some students at T2, randomly distributed in the sample. 

### 2.2. Procedure

Data were collected in classrooms with paper/pencil questionnaires during school hours in two waves, conducted by trained psychologists, researchers, and master students. The intervention took place at schools after baseline data collection.

### 2.3. KiVa Intervention

KiVa is an intensive and systematic universal school-based anti-bullying program focused on actions targeting individual children, classrooms, and schools. It was developed in the University of Turku, Finland, and it was adapted to the Italian schools in 2013–2014. It is based on the participant role model, where bullying is seen as a group process [32]. The intervention focuses on the bystanders’ reactions to a bullying situation; it aims to change their attitudes and behaviors, which assist and reinforce the bully. KiVa aims to prevent bullying and victimization and to intervene in existing bullying cases [32]. The prevention involves universal actions targeted at all students, which are delivered through 10 student lessons taught by the teachers throughout the school year. Methods used in the KiVa lessons are discussion, group work, role-play exercises, and short films about bullying, and they aim to raise awareness of the role bystanders play in the bullying process, to increase empathy toward the victim, and to provide students with safe strategies to support and defend their victimized peers. Other universal components include school posters and information for parents. Furthermore, more specific actions are targeted at students who were identified as targets or perpetrators of bullying. Both universal and indicated actions are conducted by teachers and school personnel trained in the KiVa program in two full-day face-to-face trainings and supervised in four to five meetings during school. More information about the program and the Italian adaptation can be found in Nocentini and Menesini [12].

### 2.4. Measures

The Florence Bullying–Victimization Scales (FBVS) [11] each consist of 14 items asking how often respondents experience particular behaviors either as a perpetrator or victim (e.g., “I threatened someone” for bullying, and “I was threatened” for victimization) during the past couple of months. Each item is rated on a five-point scale ranging from 1 = “never” to 5 = “several times a week”. The scales were presented after a definition of the constructs was presented. The FBVS scale showed consistent psychometric properties [11,12]. Internal reliability for both scales at T1 and T2 ranged from α = 0.82 to 0.86.

The Early Adolescent Temperament Questionnaire, Revised Short Form (EATQ-RSF) [33] was administered at T1. EATQ-RSF contains 65 questions in a self-report form asking adolescents how true each statement is for them. Response options for both forms used a five-point Likert-type scale ranging from 1 = “almost always untrue: to 5 = “almost always true”. The questionnaire measures 10 aspects of temperament (activation control, affiliation, attention, fear, frustration, high-intensity pleasure, inhibitory control, perceptual sensitivity, pleasure sensitivity, and shyness). According to the authors [34], for the present study, we used the classification based on the three superordinate dimensions of temperament: (a) effortful control (EC; based on attention, activation control, and inhibitory control), (b) negative emotionality (NE; based on fear, frustration, and shyness), and (c) positive emotionality (PE; based on surgency, pleasure sensitivity, perceptual sensitivity, and affiliation). All the subscales showed acceptable reliability coefficients: 0.71 for EC, 0.81 for NE, and 0.75 for PE.

### 2.5. Statistical Analysis

Hypotheses were tested in SPSS (SPSS Inc., Chicago, IL, USA) with linear mixed-effects models (MIXED) with full-information maximum-likelihood (ML) estimation [35]. The analysis featured a three-level (measurement occasion within individuals within schools) random-intercept model, to account for within-subject and within-school correlations. Analyses were conducted in two steps. Firstly, we tested whether EATQ temperament subscales moderated the efficacy of the KiVa program through the interaction time by group by temperament. The model tested the main effect of time, group, the three temperamental traits, and all the two- and three-level interactions. Secondly, significant moderation interactions (time by group by temperament) were followed up by creating three subgroups: top 30%, medium 40%, and bottom 30%. Subgroup analyses included the following: (a) using the same linear mixed-effect models, we tested the significance of the interaction treatment condition by time separately for subgroups of high, medium, and lower levels of temperament trait. These analyses yielded intervention effects indicating the improvement in the KiVa group relative to the control group for each subgroup defined by a different level of the temperamental traits (moderator); (b) a comparison of the effect sizes in the three subgroups separately. Effect size estimates from pre-test, post-test, and control group designs were calculated following suggestions from Morris [36]. In particular, the results from the Morris study [36] favored an effect size based on the mean pre–post change in the treatment group minus the mean pre–post change in the control group, divided by the pooled pre-test standard deviation, according to the definition of Carlson and Schmidt [37].

## 3. Results

### 3.1. Preliminary Results

Table 1 shows descriptive statistics for the outcome variables across time in both experimental and control groups. As preliminary analyses, we confirmed the finding of the previous study conducted with grade 4 and grade 6. A significant group-by-time interaction emerged for bullying (*B* = −0.014; standard error (SE) = 0.004; *p* = 0.000) and victimization (*B* = −0.024; *SE* = 0.006; *p* = 0.000), with children in the KiVa group displaying a decrease in bullying and victimization between T1 and T2, whereas children in the control group displayed a significant increase in both variables. Effect sizes estimated using all the information available from pre-test/post-test/control group designs were calculated. The estimation of effect sizes was different as compared to the previous study [12]. In the first effectiveness study, we used Cohen’s d comparing the intervention effects with the control schools at T2. Specifically, Cohen’s d was calculated as the adjusted group mean difference divided by unadjusted pooled within-group standard deviation. Results showed that, for victimization, the d pre-test/post-test/control was 0.27, and for bullying the d pre-test/post-test/control was 0.25.

### 3.2. Temperamental Moderation of Treatment Effects

Significant findings of the linear mixed models aimed at evaluating the moderation effect of temperament subscales are presented in Table 2 for both bullying and victimization outcomes. Findings showed that the trend of bullying across time is dependent on the experimental condition (KiVa vs. control) interacting with the level of effortful control and negative affectivity. Furthermore, the trend of victimization across time is dependent on the experimental condition (KiVa vs. control) interacting with the level of positive emotion.

### 3.3. Bullying Outcome: Follow-Up Analyses of High, Medium, and Low Effortful Control Levels

A significant interaction (group by time) was found in the high-EC (*B* = −0.0121; *SE* = 0.005; *p* = 0.020) and medium-EC (*B* = −0.0120; *SE* = 0.005; *p* = 0.026) subgroups, while a trend was found in the low-EC subgroup (*B* = −0.018; *SE* = 0.009; *p* = 0.060). As we can see from Table 3 and from Figure 1, the highest effect size was found in the high-EC group, where the KiVa group showed a decreasing trend over time, and the control group showed an increase over time. The same trend was found for the medium group, although with a lower effect size. The result of the low-EC subgroup was interesting; the KiVa group showed an increase over time, but this was very small as compared to the large increase in bullying over time in the low-EC participants in the control group. Thus, comparing the trend over time of KiVa and the control group within this subgroup, the effect size showed a considerable estimate, although the KiVa group did not show a significant decrease over time.

### 3.4. Bullying Outcome: Follow-Up Analyses for High, Medium, and Low Negative Emotionality

Significant interactions (group by time) were found in the low-NE (*B* = −0.016; *SE* = 0.007; *p* = 0.030) and medium-NE (*B* = −0.016; *SE* = 0.006; *p* = 0.010) subgroups, but not in the high-NE subgroup (*B* = 0.006; *SE* = 0.006; *p* = 0.379). This means that the treatment had an effect, compared to the control group, only in the low- and medium-NE subgroups. As we can see from Table 3 and Figure 2, the highest effect size was found for the low-NE subgroup, where the KiVa group showed a decreasing trend over time and the control group showed an increase over time. The same trend was found in the medium group, although with a smaller effect. Within the high-NE subgroup, the effect size comparing the trend across time of the experimental and control groups was very small, because, in both groups, these early adolescents considerably increased their level of bullying.

### 3.5. Victimization Outcome: Follow-Up Analyses for High, Medium, and Low Positive Emotionality

Significant interactions (group by time) were found in the high-PE (*B* = −0.032; *SE* = 0.011; *p* = 0.007) and medium-PE (*B* = −0.042; *SE* = 0.009; *p* = 0.000) subgroups, but not in the low-PE subgroup (*B* = −0.000; *SE* = 0.012; *p* = 0.977). This means that the treatment had an effect, compared to the control group, only in the high- and medium-PE subgroups. As we can see from Table 3 and Figure 3, the highest effect was found for the medium-PE and the high-PE subgroups, where the KiVa group showed a decreasing trend over time and the control group showed an increase over time. The effect size within the low-PE subgroup was close to 0, because, in both experimental and control groups, these young adolescents increased their level of victimization.

## 4. Discussion

The present study suggests that specific subgroups of early adolescents defined by their temperament are able to benefit most from the KiVa anti-bullying intervention, while others are more resistant to the beneficial effect of this experience. In particular, young adolescents with high levels of effortful control, and with low and medium levels of negative emotionality are the subgroups who benefit the most from the intervention in the reduction of bullying, whereas the high-NE subgroup is the most resistant. In the low-EC subgroup (a subgroup at high risk for bullying), although bullying did not decrease over time, the KiVa intervention altered the normative strong increase in bullying over time found in the control condition, with a considerable effect size. In relation to victimization, findings showed that early adolescents with high and medium levels of positive emotionality are the subgroups who benefit most from the intervention in the reduction of victimization, whereas the low-PE subgroup is the most resistant.

Findings related to the bullying outcome highlighted that the most responsive early adolescents to a universal anti-bullying intervention are those high in EC; they are able to regulate negative emotions and behaviors in the context of stressful interpersonal interactions, to employ flexible and effective coping strategies to modulate high levels of emotional reactivity, and/or to activate appropriate responses in light of changing task demands. On the other hand, the most resistant adolescents high in NE present dysregulated expression of negative affect, a high level of affective reactivity, an increased sensitivity to threat and vigilance for negative cues, and attentional bias to threatening emotional information, fear, anxiety, and threat-based reactive aggressions.

We had interesting findings in relation to the low-EC and high-NE subgroups, both potentially more at risk of bullying perpetration. The subgroup with high NE was not able to benefit from the universal anti-bullying program. On the other hand, the subgroup with low EC was affected by the program, even more than the medium-EC subgroup (see the comparison with the control group), but intervention dosage was not sufficient to decrease their level of bullying. These findings give relevant suggestions for planning selected or indicated actions. Firstly, anti-bullying programs such as KiVa are able to reduce bullying based on emotion regulation difficulties, but this is not enough for individuals with a low level of EC; for those individuals, a more intensive module on emotion regulation seems necessary. Moreover, anti-bullying programs such as KiVa are not able to reduce bullying when high levels of NE are present. The study confirmed that the NE trait is the most severe personality risk factor, because the subgroup with high NE did not respond to the intervention. This is in line with previous literature where high NE was found to be correlated with mental and physical health problems to a greater extent in comparison to other personality traits [38]. Perhaps, for these adolescents, a more intensive module which attempts to reduce high levels of NE in order to indirectly reduce risk for bullying would be more appropriate. Adaptations of cognitive and behavioral interventions, such as those developed for stress management, or to prevent anxiety disorders and depression, might be more effective in improving this dimension (e.g., for a review, see Lahey) [38]. Decreasing levels of depression and anxiety may be favorable to reduce factors triggering situations in which aggressive confrontations and bullying are likely to arise. Self-regulatory abilities may moderate the impact of NE, providing children with the capability to modulate and cope with emotionality. Increasing self-regulation may enhance children’s ability to be more flexible or competent in dealing with stressors that provoke negative emotions. 

Findings related to the victimization outcome supported a previous study where boys scoring high in environmental sensitivity benefited the most from the effects of the KiVa intervention in terms of reduced victimization scores and internalizing symptoms, but not bullying. In contrast, boys with low sensitivity did not respond to treatment at all [10]. Positive emotionality is a broad construct also involving a general sensitivity trait, and it overlaps with constructs such as extraversion and behavioral activation [39]. Thus, the findings of the present study, where the subgroup of low PE did not respond to the universal anti-bullying program, support previous findings where a specific measure of sensory-processing sensitivity was used [10].

To summarize, the subgroups most resistant to the KiVa universal anti-bullying intervention are high NE for bullying—but not low EC—and low PE for victimization. One possible explanation is the fact that the KiVa program addressed the vulnerability of the low-EC adolescents’ profile through specific activities, but not the vulnerability of the high-NE and low-PE subgroups. More specific components covering these aspects are needed if we want anti-bullying interventions to be able to address bullying and victimization in these high-risk subgroups of early adolescents as well. 

### Limitations and Future Studies

The current study presents some limitations. Firstly, we relied on self-report data for the evaluation of bullying and victimization and of temperament, which raises the probability of reporter bias. Secondly, the effect sizes in the current study ranged from 0.09 to 0.30, representing a small effect size. However, we need to specify that these are the effects of a universal program on subgroups. The effect sizes are typically smaller in universal prevention studies than in more indicated interventions, and the effect sizes in effectiveness studies are also typically smaller than in efficacy studies [4]. Thirdly, other relevant moderators should be considered in a more articulated design assessing the moderation effect of different types of variables, such as socio-demographic, temperamental, individual, and contextual variables. Fourthly, the findings of the current study could be specific for Italian culture and cannot be generalized to other countries. Further research should also be focused on interesting research questions such as whether temperament interacts with the “pure condition” (bully or victim) or the double role (bully/victim). Moreover, this study evaluated the moderation of temperament on the effectiveness of a universal anti-bullying program. Finally, future studies could evaluate whether temperament is also able to moderate the effectiveness of selective and indicated actions devoted to the decrease of bullying and victimization, contributing to Tier 2 and Tier 3 as well.

## 5. Conclusions

Despite these limitations, the present study moved beyond merely examining the average treatment effect by identifying which part of the distribution is most affected by the intervention, and by giving suggestions for future targeting and program roll-out. The analyses used in this study were able to account for the nested design, using a subject and school random effect. Furthermore, the test of moderation effects in a single model without conducting multiple moderation tests, as well as the subgroups analyses conducted following the most recent literature [17,30], represents a strength of the current study.

The identification of non-responders to the universal anti-bullying interventions has relevant implications because of their impact on Tier 2 and Tier 3 interventions. Overall, the current study showed that high-NE and low-PE early adolescents are more resistant to the intervention. Maybe these subgroups need a longer intervention or an intervention with a more intense dosage or with a different component (Tier 2 and Tier 3). The higher the number of students responding to the universal (Tier 1) preventive interventions is, the lower the number of students to be involved in advanced tier supports (Tier 2 and Tier 3) will be. Given that students involved in Tier 2 and Tier 3 for bullying prevention are at increased risk for future psychological, social, and school consequences, providing better universal prevention that meets the needs of high-risk early adolescents is crucial.

## Figures and Tables

**Figure 1 ijerph-16-00388-f001:**
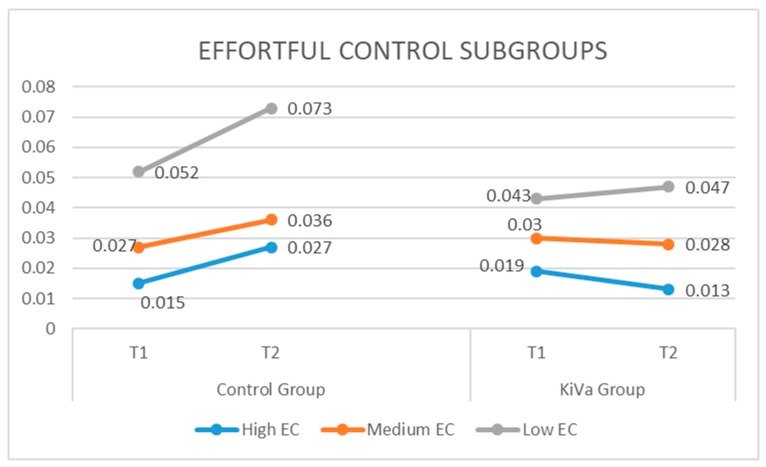
Bullying across time distinguishing between control and KiVa groups, and high, medium, and low levels of effortful control.

**Figure 2 ijerph-16-00388-f002:**
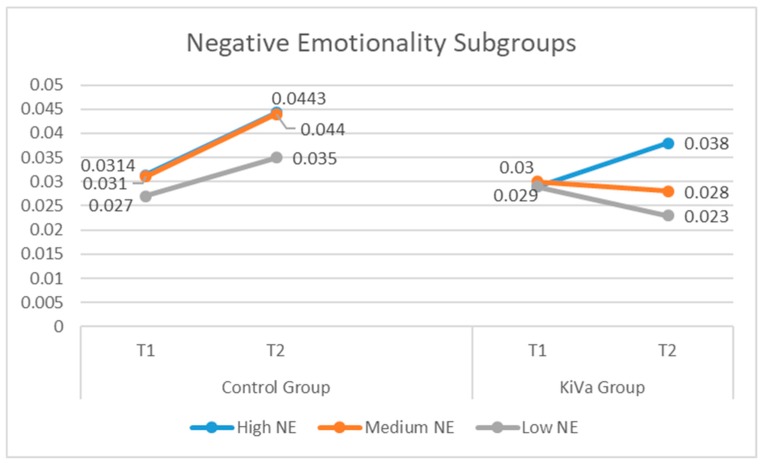
Bullying across time distinguishing between control and KiVa groups, and high, medium, and low levels of negative emotionality.

**Figure 3 ijerph-16-00388-f003:**
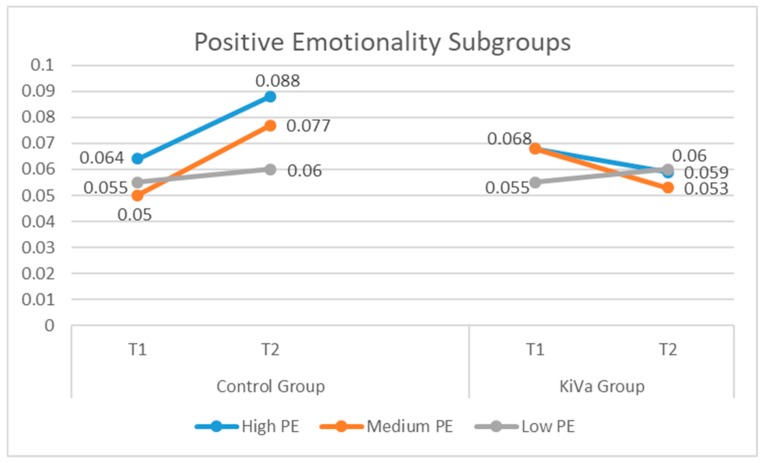
Victimization across time distinguishing between control and KiVa groups, and high, medium, and low levels of positive emotionality.

**Table 1 ijerph-16-00388-t001:** Descriptive statistics for the behavioral outcomes: means (M) and standard deviations (SD).

Outcomes	Data Collection	Middle School	
		Experimental M (SD)	Control M (SD)
Victimization	T1 (*n* = 1049)	*n* = 533; 0.062 (0.096)	*n* = 516; 0.056 (0.080)
	T2 (*n* = 987)	*n* = 494; 0.057 (0.073)	*n* = 493; 0.075 (0.086)
Bullying	T1 (*n* = 1045)	*n* = 529; 0.032 (0.059)	*n* = 516; 0.030 (0.050)
	T2 (*n* = 986)	*n* = 493; 0.029 (0.053)	*n* = 493; 0.041 (0.063)

Notes: Differences in the number of subjects within the same time are due to missing data in the specific variable. Means and standard deviations are based on subjects with full information available used in the analysis.

**Table 2 ijerph-16-00388-t002:** Mixed model predicting the moderating role of effortful control (EC), negative emotionality (NE), and positive emotionality (PE) on bullying and victimization outcomes.

Predictor	Outcome: Bullying		Outcome: Victimization		
	*B* (SE)	*p*	*B* (SE)		*p*
Intercept	**0.093 (0.022)**	**0.000**	**0.065 (0.032)**		**0.040**
Time	−0.018 (0.025)	0.445	0.029 (0.037)		0.424
Group	**0.087 (0.032)**	**0.006**	−0.034 (0.045)		0.449
EC	**−0.007 (0.002)**	**0.000**	**−0.005 (0.002)**		**0.028**
NE	0.001 (0.001)	0.202	**0.005 (0.002)**		**0.002**
PE	0.000 (0.001)	0.772	−0.001 (0.002)		0.466
Time × group	**−0.062 (0.035)**	**0.073**	−0.014 (0.053)		0.791
EC × time	0.000 (0.002)	0.741	−0.002 (0.003)		0.460
NE × time	**−0.003 (0.001)**	**0.010**	−0.002 (0.002)		0.233
PE × time	**0.003(0.001)**	**0.006**	0.001 (0.002)		0.516
EC × group	**−0.006(0.002)**	**0.004**	−0.001(0.003)		0.712
NE × group	**−0.003 (0.001)**	**0.010**	0.000 (0.003)		0.830
PE × group	0.002(0.002)	0.218	**0.005(0.002)**		**0.050**
**Time** × **EC** × **group**	**0.005 (0.002)**	**0.039**	0.006 (0.004)		0.080
**Time** × **NE** × **group**	**0.004 (0.002)**	**0.046**	−0.002 (0.003)		0.611
**Time** × **PE** × **group**	−0.003 (0.002)	0.127	**−0.005 (0.002)**		**0.049**
Residual variance	**0.002 (0.000)**	**0.000**	**0.004 (0.000)**		**0.000**
Subjects: random intercept	**0.001 (0.000)**	**0.000**	**0.002 (0.000)**		**0.000**
Schools: random intercept	0.000 (0.000)	0.126	0.000 (0.000)		0.188

Note: Boldface type indicates statistically significant results (*p* < 0.05) from the deviance tests for fixed effects and from the Wald tests for random effects.

**Table 3 ijerph-16-00388-t003:** Effect sizes (Cohen’s *d*) estimates from pre-test/post-test/control group designs in each temperamental subgroup.

Subgroup	*d* Pre-Test/Post-Test/Control
High effortful control	0.30
Medium effortful control	0.22
Low effortful control	0.24
High negative emotionality	0.09
Medium negative emotionality	0.28
Low negative emotionality	0.30
High positive emotionality	0.35
Medium positive emotionality	0.48
Low positive emotionality	0.01

Notes: The effect size is based on the mean pre–post change in the treatment group minus the mean pre–post change in the control group, divided by the pooled pre-test standard deviation (see Morris, 2008).

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
