# Peer review of "For Whom Is Anti-Bullying Intervention Most Effective? The Role of Temperament"

_ijerph, 2019, doi:10.3390/ijerph16030388_

Round 1
Reviewer 1 Report
This manuscript presents the findings of a study of whether temperment of young adolescents moderates the effects of the KiVa anti-bullying intervention. The study goes beyond efficacy to evaluate the moderators of temperment and can be a predictor of who might be most successful in this KiVa program.
Overall, the manuscript is well written and adds a valuable contribution to the literature. The background is well referenced and supports the study. It would be helpful if more information about the KiVa intervention was presented earlier in the background to set the context more clearly to those unfamiliar with the program and its' components. In the methods section, KiVa is briefly presented, and I felt more information here too would be helpful. The measures and data analytic plan were sound and the results were well presented and explained in discussion. The conclusions do not overreach.
Author Response
1. It would be helpful if more information about the KiVa intervention was presented earlier in the background to set the context more clearly to those unfamiliar with the program and its' components. In the methods section, KiVa is briefly presented, and I felt more information here too would be helpful.
The background is focused on the more general topic of the paper, which is the moderation of effectiveness by individual variables. Thus, we think that it is not appropriate to extend the introduction to add information about KiVa. However, we presented more information in the method section.
Reviewer 2 Report
For whom is anti-bullying intervention most effective? The role of temperament
Many thanks to the authors and the editor for allowing me to evaluate this interesting work.
This paper describes a study investigating whether early adolescents’ temperament Effortful Control (EC), Negative Emotionality (NE) and Positive Emotionality (PE) moderates the effects of the KiVa anti-bullying program..
The article is properly written, following all the guidelines for publications of scientific articles, the objective is to present the evaluation of the impact of KiVa anti-bullying program varied as a function of early adolescents’ temperament defined using the three superordinate dimensions: Effortful Control (EC), Negative Emotionality (NE) and Positive Emotionality (PE).
Methods
The method is perfectly described by presenting the sociodemographic variables, the place, and the way of performing the simulations, and how everything done through the corresponding statistical tests was evaluated later.
The article has gone through an ethical committee having its approval.
Results
Note that the results respond to the objective set at the beginning, through a sample of 1,052 participants who responded to the pre-intervention and 984 participants who responded the subsequent survey.
The tables and data presented are correct.
Discussion
The discussion is perfectly organized, and if perhaps you missed some discussion with studies in other countries, not only comparing with a single previous study. I would have liked to see the international panorama regarding this topic.
Conclusions
The conclusions should be limited to the country of study, and not generalized, since it is due to their part very well due to the great differences at a level of health between some countries and others, but it can not stop being good answers to the authors modify that part
Author Response
1. The discussion is perfectly organized, and if perhaps you missed some discussion with studies in other countries, not only comparing with a single previous study. I would have liked to see the international panorama regarding this topic.
In agreement to this suggestion I added and specified the country in each of the study presented in the introduction related to the association between personality and bullying-victimization.
2. The conclusions should be limited to the country of study, and not generalized, since it is due to their part very well due to the great differences at a level of health between some countries and others, but it can not stop being good answers to the authors modify that part
In agreement to this suggestion, we added this issue in the discussion section.
Reviewer 3 Report
This is, in summary, a study aimed to investigate whether early adolescents’ temperament - Effortful Control (EC), Negative Emotionality (NE) and Positive Emotionality (PE) - moderates the effects of the KiVa anti-bullying program in a sample of 1051 sixth-grade early adolescents, randomly assigned to the KiVa intervention or the control condition. Overall, the authors reported that EC and NE moderated intervention effects on bullying, indicating that subgroups with high levels of EC, and low and medium levels of NE are those who benefited most from the intervention. When compared to the control condition, the Low EC subgroup showed a lower increase, with a considerable effect size. Conversely, the high NE subgroup did not show any positive effects compared to the control group. Concerning victimization, it has been reported that early adolescents with high and medium levels of PE are the subgroups who benefit most from the intervention whereas the low PE subgroup is the most resistant group. The authors concluded about the importance of considering temperament as a moderator of intervention effects, since interventions tailored to early adolescents with specific traits might yield larger effects.
The authors may find as follows my main comments/suggestions.
First, i suggest to generally remove the term “temperaments” from the whole text, replacing it with the term “personality traits”. Subthreshold affective temperaments traits commonly measured using specific psychometric instruments such as the TEMPS-A (Akiskal and Akiskal, 2005), have been conceptualized in psychiatric settings as biological-derived factors playing a significant role in the psychopathological characteristics of mood disorders including the clinical evolution of minor/major mood episodes, the direction of polarity, the clinical symptomatology, the long-term course, suicidality and even medication adherence, while personality traits are generally hypothesized as sufficiently stable traits that are usually present even before the onset of a psychiatric disorder or a medical disease.
Also, how environmental sensitivity has been conceptualized within the main section is unclear and needs to be specified in a more detailed manner.
Moreover, as the authors focused on adolescemce, they could also mention the consequences related to having experienced specific adverse life events (e.g., bullying or cyberbullying behaviors) and/or negative outcomes in young children/adolescents. Based on the main results of a recent systematic review, the number of experienced adversities or negative life events seemed to have an important effect on negative outcome in youth individuals. In order to briefly focus on the specified association (although i understand that the link adverse life events and negative outcomes in young people is not the main topic of this paper), i suggest to cite and discuss some specific studies in this field.
Furthermore, the main reasons why from 1052 early adolescents of grade 6, only 984 filled out the questionnaires at T2 need to be reported extensively by the authors.
In addition, the description of the florence bullying-victimization scales as well as the early adolescent temperament questionnaire may be more succinct.
The relevance of NE could to be better stressed throughout the main text. Although the authors stated that anti-bullying programs such as KiVa are not able to reduce bullying when high levels of NE are present as NE trait is the most severe personality risk factor, whether possible resilient factors against negative emotionality exist might be reported. This could be of great interest for the readers,
Finally, what is the take-home message of this manuscript? While, the authors reported that high NE and low PE early adolescents are more resistant to the intervention, they should even report how these findings may be translated into the clinical practice.
Author Response
1. First, i suggest to generally remove the term “temperaments” from the whole text, replacing it with the term “personality traits”. Subthreshold affective temperaments traits commonly measured using specific psychometric instruments such as the TEMPS-A (Akiskal and Akiskal, 2005), have been conceptualized in psychiatric settings as biological-derived factors playing a significant role in the psychopathological characteristics of mood disorders including the clinical evolution of minor/major mood episodes, the direction of polarity, the clinical symptomatology, the long-term course, suicidality and even medication adherence, while personality traits are generally hypothesized as sufficiently stable traits that are usually present even before the onset of a psychiatric disorder or a medical disease.
Thank you to the reviewer for this interesting point. However, we think that we need to use the term temperament because we have measured this construct using the Early Adolescence Temperament Questionnaire by Rothbart and colleagues, which is a international validated instrument measuring Temperament and not Personality traits. Besides, in developmental psychology the most important measure of individual traits are temperamental characteristics.
2. Also, how environmental sensitivity has been conceptualized within the main section is unclear and needs to be specified in a more detailed manner.
In agreement to this suggestion, we added a definition at page 2.
3. Moreover, as the authors focused on adolescemce, they could also mention the consequences related to having experienced specific adverse life events (e.g., bullying or cyberbullying behaviors) and/or negative outcomes in young children/adolescents. Based on the main results of a recent systematic review, the number of experienced adversities or negative life events seemed to have an important effect on negative outcome in youth individuals. In order to briefly focus on the specified association (although i understand that the link adverse life events and negative outcomes in young people is not the main topic of this paper), i suggest to cite and discuss some specific studies in this field.
Thank you to the reviewer, this is an interesting point. However, in our view, this is not the point of the current paper. We have already specified in different parts of the manuscript which are the consequences of the victimization.
4. Furthermore, the main reasons why from 1052 early adolescents of grade 6, only 984 filled out the questionnaires at T2 need to be reported extensively by the authors.
We specified that poi at page 3: The attrition is mainly explained by a daily absence of the students at T2 randomly distributed in the sample.
5. In addition, the description of the florence bullying-victimization scales as well as the early adolescent temperament questionnaire may be more succinct.
In agreement to this point we deleted some parts in these parts.
6. The relevance of NE could to be better stressed throughout the main text. Although the authors stated that anti-bullying programs such as KiVa are not able to reduce bullying when high levels of NE are present as NE trait is the most severe personality risk factor, whether possible resilient factors against negative emotionality exist might be reported. This could be of great interest for the readers,
In agreement to this suggestion, we added a discussion at page 11.
7. Finally, what is the take-home message of this manuscript? While, the authors reported that high NE and low PE early adolescents are more resistant to the intervention, they should even report how these findings may be translated into the clinical practice.
We specified in different parts at page 11-12 the implications of these findings and how the results can suggest specific information on how to improve the specific actions.
See for example:
…..Perhaps, for these adolescents, a more intensive module which attempts to reduce high levels of NE in order to indirectly reduce risk for bullying would be more appropriate. Adaptations of cognitive and behavioral interventions, such as those developed for stress management, or to prevent anxiety disorders and depression, might be more effective in improving this dimension (e.g., for review see Lahey) [38]. Decreasing levels of depression and anxiety may be favorable to reduce factors triggering situations in which aggressive confrontations and bullying are likely to arise.
…….. More specific components covering these aspects are needed if we want anti-bullying interventions to be able to address bullying and victimization in these high risk subgroups of early adolescents as well.
……… more resistant to the intervention. Maybe these subgroups need a longer intervention or an intervention with a more intense dosage or with a different component (Tier 2 and Tier 3). The higher the number of students responding to the universal (Tier 1) preventive interventions is, the lower the number of students to be involved in advanced tier supports (Tier 2 and Tier 3) will be. Given that students involved in Tier 2 and Tier 3 for bullying prevention are at increased risk for future psychological, social and school consequences, providing better universal prevention that meets the needs of high-risk early adolescents is crucial.